# Comprehensive Genome-Wide Analysis and Expression Pattern Profiling of *PLATZ* Gene Family Members in *Solanum Lycopersicum* L. under Multiple Abiotic Stresses

**DOI:** 10.3390/plants11223112

**Published:** 2022-11-15

**Authors:** Antt Htet Wai, Md Mustafizur Rahman, Muhammad Waseem, Lae-Hyeon Cho, Aung Htay Naing, Jong-Seong Jeon, Do-jin Lee, Chang-Kil Kim, Mi-Young Chung

**Affiliations:** 1Department of Agricultural Education, Sunchon National University, 413 Jungangno, Suncheon 57922, Republic of Korea; 2Department of Biology, Yangon University of Education, Kamayut Township 11041, Yangon Region, Myanmar; 3Graduate School of Biotechnology and Crop Biotech Institute, Kyung Hee University, Yongin 17104, Republic of Korea; 4Department of Botany, University of Narowal, Narowal 51600, Pakistan; 5Department of Plant Bioscience, College of Natural Resources and Life Science, Pusan National University, Miryang-si 50463, Gyeongsangnam-do, Republic of Korea; 6Department of Horticulture, Kyungpook National University, Daegu 41566, Republic of Korea

**Keywords:** PLATZ, tomato, transcription factors, abiotic stress, expression analysis, subcellular localization, gene co-expression network

## Abstract

PLATZ (plant AT-rich sequence and zinc-binding) family proteins with two conserved zinc-dependent DNA-binding motifs are transcription factors specific to the plant kingdom. The functions of PLATZ proteins in growth, development, and adaptation to multiple abiotic stresses have been investigated in various plant species, but their role in tomato has not been explored yet. In the present work, 20 non-redundant *Solanum lycopersicum PLATZ* (*SlPLATZ*) genes with three segmentally duplicated gene pairs and four tandemly duplicated gene pairs were identified on eight tomato chromosomes. The comparative modeling and gene ontology (GO) annotations of tomato PLATZ proteins indicated their probable roles in defense response, transcriptional regulation, and protein metabolic processes as well as their binding affinity for various ligands, including nucleic acids, peptides, and zinc. *SlPLATZ10* and *SlPLATZ17* were only expressed in 1 cm fruits and flowers, respectively, indicating their preferential involvement in the development of these organs. The expression of *SlPLATZ1*, *SlPLATZ12*, and *SlPLATZ19* was up- or down-regulated following exposure to various abiotic stresses, whereas that of *SlPLATZ11* was induced under temperature stresses (i.e., cold and heat stress), revealing their probable function in the abiotic stress tolerance of tomato. Weighted gene co-expression network analysis corroborated the aforementioned findings by spotlighting the co-expression of several stress-associated genes with *SlPLATZ* genes. Confocal fluorescence microscopy revealed the localization of SlPLATZ–GFP fusion proteins in the nucleus, hinting at their functions as transcription factors. These findings provide a foundation for a better understanding of the structure and function of *PLATZ* genes and should assist in the selection of potential candidate genes involved in the development and abiotic stress adaptation in tomato.

## 1. Introduction

A transcription factor (TF) is a regulatory protein with one or more DNA-binding domains that can control the expression of target genes by binding directly to their promoter regions. From a combinatorial gene regulation perspective, TFs can also physically interact with each other to control diverse cellular processes [1]. The evolution of the TF families does not always demonstrate a robust corresponding correlation among plants, animals, and yeast. In the dicotyledonous model plant Arabidopsis, more than 5% of its genome codes for over 1500 transcription factors, nearly 45% of which belong to plant-specific TF families [2]. Some classes of transcription factors identified more abundantly in plants (i.e., WRKY, NAC, and AP2/EREBP families) display their critical roles in plant-specific mechanisms such as secondary metabolism, responses to plant hormones, and the development of unique characteristics of distinct cell types [3,4,5,6].

Plant AT-rich sequence and zinc binding (PLATZ) proteins, identified as a novel class of zinc-dependent DNA-binding proteins, belong to the plant-specific TF family. They maintain the distinctive structural conformation of the zinc finger protein family and harbor two evolutionarily conserved domains required for zinc-dependent DNA binding, namely C-x_2_-H-x_11_-C-x_2_-C-x_(4–5)_-C-x_2_-C-x_(3–7)_-H-x_2_-H and C-x_2_-C-x_(10–11)_-C-x_3_-C [7]. The two conserved regions of *PLATZ* genes identified in a variety of plant species differ from the zinc-binding motifs of other zinc finger proteins, such as COSTANS/CONSTANS-like (CO/COLs), GATA finger (C-x_2_-C-x_(17–18)_-C-x_2_-C), the DNA-binding finger (Dof), LIM (C_2_HC_5_), and RING (C_3_HC_4_) [8,9,10,11]. These zinc finger proteins, which have been categorized into several different types according to the quantity and arrangement of their cysteine and histidine residues that bind the zinc ion [12], regulate various cellular functions, including binding of RNA, regulation of apoptosis, transcriptional control, and interaction among proteins [13]. Additionally, their functions as key transcriptional regulators implicated in the defense and acclimation responses of various plant species have been well documented [13,14]. Since the first *PLATZ* gene member, *PsPLATZ1*, was identified from pea (*Pisum sativum*) in 2001 as a transcriptional inhibitor that binds to A/T-rich regions [7], it has aroused the interest of researchers to explore the functions of PLATZ transcription factors in diverse plant species.

PLATZ transcription factors regulate the growth and development of numerous plant species. In rice, the PLATZ transcription factor GL6 interacts with transcription factor class C1 and RNA polymerase III subunit C53 to control spikelet number and grain length [15], while the PLATZ protein SHORT GRAIN6 (SG6) modulates seed size and the division of spikelet hull cells by interacting with DP transcription factor and cell division regulators [16]. *Floury3* (*FL3*), encoding a PLATZ transcription factor in maize, regulates endosperm development and the accumulation of storage materials in maize grains through the interaction with transcription factor class C1 (TFC1) and RNA polymerase III subunit 53 (RPC53) [17]. Another maize PLATZ transcription factor, ZmPLATZ2, also functions in the regulation of starch synthesis in maize by directly binding to the CAAAAAAA motif in the promoter region of *ZmSSI* [18]. The important roles of PLATZ proteins in the secondary growth of *Populus* stems were demonstrated through transcriptome analysis [19]. The PLATZ transcription factor ORESARA15 (ORE15) controls leaf development and senescence in Arabidopsis by modulating the rate and duration of early cell proliferation and the GRF/GIF regulatory pathway [20], whereas ABA-INDUCED expression 1 (AIN1), a putative PLATZ protein family member in Arabidopsis, affects the elongation of the primary root in response to ABA induction [21]. It has been recently reported that ORE15 also regulates root meristem enlargement in Arabidopsis via auxin and cytokinin signaling-related pathways [22].

In addition to their involvement in the growth and developmental processes of plants, PLATZ transcription factors also play a crucial role in plant acclimatization to various environmental stimuli. *AtPLATZ1* and *AtPLATZ2* are reported to play a regulatory role in desiccation tolerance in vegetative tissues and seeds of Arabidopsis [23]. The expression of *PhePLATZ1* is induced by osmotic stress, and the *PhePLATZ1*-overexpressing transgenic Arabidopsis lines show higher tolerance to drought stress compared to wild-type plants [24]. Furthermore, AtPLATZ4 functions as a positive regulator in ABA response and drought tolerance in Arabidopsis by directly interacting with the A/T-rich sequences in the promoter of *PIP2;8* [25]. By contrast, *GmPLATZ17* plays a negative regulatory role in the drought tolerance of soybean by suppressing the transcription of *GmDREB5* [26]. *AtPLATZ2* functions redundantly with *AtPLATZ7* as a negative regulator in the salt stress adaptation of Arabidopsis by repressing the expression of *CBL4*/*SOS3* and *CBL10/SCaBP8* [27]. The overexpression of cotton *PLATZ1* in Arabidopsis enhances the osmotic and salt stress tolerance of the resulting transgenic lines at the germination and seedling stages [28], while that of soybean *PLATZ1* in Arabidopsis shows delayed germination under osmotic stress [29]. Furthermore, several reports have demonstrated that *PLATZ* genes in multiple plant species are responsive to various hormones and abiotic stresses [28,30,31,32].

Tomato (*Solanum lycopersicum* L.) is of commercial value due to its widespread production and consumption around the globe. As a result of the abundance of several nutrients, such as lycopene, lutein, Zeaxanthin, potassium, ascorbic acid, and β-carotene in tomato, it is also the second most significant vegetable crop in the world [33]. With the availability of whole genome sequences of tomato cultivars as well as with its numerous interesting features, such as compound leaves, a sympodial shoot, and fleshy fruit, which cannot be analyzed in other model plants (i.e., Arabidopsis and rice), tomato has also been extensively studied as a model organism for fleshy-fruited plants [34]. Genome-wide analyses utilize various bioinformatics tools to explore the characteristics of gene family members, gene structures, phylogenetic relationships, expression profiles, gene ontologies, putative gene functions, and so forth. The data extracted from genome-wide identification studies provide researchers with valuable information and an initial framework for further functional verification of potential candidate genes that can be exploited for plant genetic improvement [35]. A genome-wide characterization of the *PLATZ* gene family in diverse plant species has been conducted [14,32,36,37,38,39,40], but there has been no systematic investigation of their phylogenetic relationships or characteristics in solanaceous crops. The productivity and fruit quality of tomato are severely affected by various adverse environmental stresses, and hence it is essential to identify the potential novel genes implicated in the abiotic stress responses of tomato for the utilization of these genes in the genetic improvement of tomato [41]. Therefore, we conducted expression profiling of tomato *PLATZ* genes at various developmental stages and on exposure to diverse abiotic stresses. In addition to the genome-wide characterization of SlPLATZ proteins, their subcellular localization assays and gene co-expression network analysis using RNA sequencing data were carried out to shed light on their probable functional role in tomato. This study aimed to provide a solid foundation for further functional verification of potential PLATZ transcription factors for the improvement of abiotic stress adaptation in tomato.

## 2. Results

### 2.1. Identification and Sequence Analysis of PLATZ Family Members in Tomato

A total of 20 non-redundant *PLATZ* genes were identified in the tomato genome and designated as *SlPLATZ1***–***SlPLATZ20* in accordance with their chromosomal positions. In silico analyses revealed that the lengths of predicted PLATZ proteins varied from 137 (SlPLATZ6) to 255 (SlPLATZ14) amino acids (aa), and their molecular weight ranged between 15.94 and 29.16 kDa for SlPLATZ6 and SlPLATZ14, respectively. The theoretical pI value of only one protein (SlPLATZ6) was less than 7, whereas those of the other 19 proteins were greater than 7, revealing that all tomato PLATZ family proteins were basic proteins, except SlPLATZ6. The GRAVY of SlPLATZ proteins ranged between −0.217 and −0.709, showing that all these proteins were hydrophilic. Further information on SlPLATZ family proteins, including locus IDs, is shown in Table 1.

### 2.2. Multiple Protein Sequence Alignment and Phylogenetic Analysis

Multiple sequence alignment of tomato PLATZ proteins with a PLATZ protein from the model plant Arabidopsis underlined the evolutionary conservation of PLATZ proteins from tomato and Arabidopsis (Figure 1), particularly the presence of the two conserved regions, the first region harboring five cysteine and three histidine residues [C-x_2_-H-x_11_-C-x_2_-C-x(_4–5_)-C-x_2_-C-x(_3–5_)-H-x_2_-H] and the second region with four cysteine residues [C- x_2_-C-x(_10–12_)-C-x_3_-C], crucial for zinc-dependent DNA binding [38,39]. However, it is interesting to note that SlPLATZ6 is the only protein devoid of the second conserved region. To investigate the evolutionary relationship of the *SlPLATZ* gene family, a phylogenetic tree was constructed using PLATZ proteins from 10 different plant species. (Figure 2). PLATZ family members from various plant species were divided into two primary clades, which can be further categorized into seven groups based on their phylogenetic affinity. The evolutionary tree constructed indicated a major grouping of homologous proteins from monocots (maize, rice, and sorghum) and dicots (*Brassica rapa*, tomato, Arabidopsis, and potato) together with lower plants. Intriguingly, PLATZ proteins from the green alga *Chlamydomonas reinhardtii* were distributed specifically in groups 2 and 3 of clade I and clustered with the homologs from monocots, dicots, and the basal angiosperm *Amborella trichopoda*. The paralogous proteins from the moss *Physcomitrella patens* were clustered together in groups 1 and 3, but PpPLATZ13 was paired with the homolog from *A. trichopoda* in group 6. The basal angiosperm homologs, distributed in groups 1, 3, 5, 6, and 7, were clustered not only with those from dicots and monocots but also with those from the lower plants. The 20 tomato PLATZ proteins were allocated to four of the seven groups, except groups 3, 5, and 6. The greatest number (12) of SlPLATZs was arranged in group 2, while the smallest number (1) was in group 4. Groups 1 and 7 included four and three SlPLATZ proteins, respectively. PLATZ proteins from tomato were grouped specifically with those of their evolutionarily closely related species, potato.

### 2.3. Conserved Domain Prediction, Gene Structure, and Motif Composition of SlPLATZ Genes

PLATZ proteins are composed of a conserved PLATZ domain with or without additional types of domains. Seventeen out of 20 SlPLATZ proteins comprised only the PLATZ domain, while the remaining three SlPLATZ proteins were composed of the PLATZ domain and an additional domain, viz., the B-box-type zinc finger domain (BBOX) or transmembrane domain (TM) (Figure 3). Among the three proteins having an additional domain, SlPLATZ4 was the only member harboring a TM domain in the rear of the PLATZ domain, while a BBOX domain positioned before the PLATZ domain was predicted in SlPLATZ15 and SlPLATZ16. Analysis of the exon–intron distribution of the *SlPLATZ* genes indicated that there was no gene lacking an intron in their coding sequences (Appendix A). Except for *SlPLATZ5*, which contained only one intron, all *SlPLATZ* genes have between two and three introns. Intriguingly, the paralogous gene pairs in the same clades, such as *SlPLATZ15* and *SlPLATZ16*, shared the identical gene structure in terms of intron number and exon length.

To further analyze the structural diversity of the *PLATZ* gene family, we explored the composition of conserved motifs in SlPLATZ proteins with respect to the PLATZ homologous proteins from Arabidopsis and rice (Appendix A). The paralogous gene pairs and orthologous gene pairs from the three plant species harbored almost identical motif compositions. The motif organization of most SlPLATZ proteins belonging to phylogenetic group 2 is almost identical, with the exception of SlPLATZ5, which contained only motifs 1, 3, and 6. Motifs 1 and 5, which resided in the PLATZ domain region, were the most prevalent motifs in nearly all the PLATZ homologs of three plant species, except for AtPLATZ9 and OsPLATZ15, which were devoid of motif 1 and 5, respectively. Intriguingly, motif 7 was uniquely present in most SlPLATZ protein members belonging to group 2, but not identified in any PLATZ homologs from rice and Arabidopsis. In addition, motif 9 was specific to certain PLATZ proteins from all plant species belonging to group 1.

### 2.4. Chromosomal Distribution, Gene Duplication, and Microsynteny Analysis of SlPLATZ Genes

*SlPLATZ* genes were unequally dispersed on eight of the twelve chromosomes (Chr), except for Chr03, Chr05, Chr09, and Chr12 (Appendix A). The majority of tomato chromosomes harbored a small number of *PLATZ* genes (1–3) residing near the distal end of the chromosomes, while Chr02 possessed 10 genes located in the central region of the chromosome. Of the chromosomes with a lesser number of *PLATZ* genes, Chr07 had three genes, which was followed by Chr08 with two genes. The remaining five chromosomes (Chr01, Chr04, Chr06, Chr10, and Chr11) carried only one gene each.

A total of seven duplicated gene pairs were identified in the tomato *PLATZ* gene family (Table 2, Appendix A). Among the 20 *SlPLATZ* genes, four gene pairs (*SlPLATZ2*/*SlPLATZ3*, *SlPLATZ3*/*SlPLATZ4*, *SlPLATZ6*/*SlPLATZ7*, and *SlPLATZ15*/*SlPLATZ16*) were predicted to be tandemly duplicated, while three gene pairs (*SlPLATZ13*/*SlPLATZ18*, *SlPLATZ14*/*SlPLATZ20*, and *SlPLATZ17*/*SlPLATZ18*) were detected as segmental duplicates. The duplicated genes of each pair were located in the same phylogenetic group. The mode of selection pressure imposed on the *SlPLATZ* gene duplicates was identified based on Ka/Ks values. The Ka/Ks ratios of all *SlPLATZ* gene duplicates were lower than 1, revealing that these genes had undergone strong purifying/negative selection with little variation after duplication. The computation of the Ka/Ks ratios of gene duplicates indicated that the duplication events were predicted to have occurred between 1.05 and 31.43 million years ago (MYA). These duplicated gene pairs may have diverged in the last 1–31 million years.

Furthermore, a comparative microsyntenic analysis was performed to investigate the phylogenetic relationship between tomato, Arabidopsis, and rice. No orthologous gene pairs were predicted between rice and Arabidopsis, whereas we identified seven orthologs between tomato and Arabidopsis and two orthologs between tomato and rice, respectively (Figure 4). *SlPLATZ18* was orthologous to three *PLATZ* genes (*AtPLATZ3*, *AtPLATZ11,* and *AtPLATZ12*) from Arabidopsis, while *SlPLATZ12* and *SlPLATZ17* were orthologous to *AtPLATZ6* and *AtPLATZ11*, respectively. Intriguingly, *SlPLATZ1* and *SlPLATZ11* were collinear with the *PLATZ* genes from both Arabidopsis and rice. *SlPLATZ1* and *SlPLATZ11* were orthologous to *AtPLATZ10* and *OsPLATZ6*, and *OsPLATZ11* and *SlPLATZ2*, respectively.

### 2.5. Analysis of Cis-Acting Elements and miRNA Target Sites in SlPLATZ Genes

We analyzed the *cis*-regulatory elements in the upstream regions of *SlPLATZ* gene family members to explore their probable involvement in the transcriptional regulation on exposure to diverse abiotic stresses. Different numbers of *cis*-acting elements associated with phytohormones and abiotic stress adaptation were predicted in the promoter sequences of tomato *PLATZ* genes (Appendix A and Appendix A). The identified stress-responsive *cis*-elements were LTR (related to low temperature and hypersalinity stress and defense), TC-rich repeats (involved in defense and stress responses), drought-responsive MYB-binding site (MBS), and WUN motifs (implicated in wounding responses), whereas those associated with hormones were TGA elements and AuxRR-core (related to the auxin response), ABA-responsive ABRE elements, TGA elements and AuxRR-core (with roles in the auxin response), TCA-elements (implicated in the SA response), the TGACG- and CGTCA-motifs (involved in the jasmonic acid [JA] response), and gibberellic acid (GA) response-related TATC- and P-boxes.

Transcription factors (TFs) and microRNAs (miRNAs) are crucial gene regulatory factors, and miRNA-mediated regulation of transcription factors plays a vital role in plant development and stress tolerance [42]. Thus, we examined the miRNA target sequences in *SlPLATZ* genes to shed light on the relationship between miRNA and tomato *PLATZ* transcription factors in environmental stress response. A total of 12 out of 20 tomato *PLATZ* genes were predicted to be the target genes of diverse tomato miRNAs, including *sly-miR159*, *sly-miR162*, *sly-miR171a*, *sly-miR319a*, *sly-miR319b*, *sly-miR319c-3p*, *sly-miR396a-5p*, *sly-miR396b*, *sly-miR403-5p*, *sly-miR482a*, *sly-miR5303*, *sly-miR6024*, *sly-miR9469-3p*, *sly-miR9472-3p,* and *sly-miR9478-3p*, which are involved in the development and stress response of tomato (Appendix A).

### 2.6. Analysis of Three-Dimensional Structures of Predicted SlPLATZ Proteins

The 3D model structures of tomato PLATZ proteins predicted through the I-TASSER server showed creditable values of parameters used in comparative modeling (Figure 5), including the percentage of sequence identity, coverage, and Z-scores (Appendix A). This indicated the reliability of the predicted SlPLATZ models. The binding sites in the resulting 3D models were determined with Discovery Studio v.21.1, and the secondary structural components predicted in the models ranged from 2–8 for α-helices, 5–8 for β-strands, and 8–15 for coils, respectively. The majority of duplicated gene pairs in the *SlPLATZ* gene family displayed similar secondary structural components, but SlPLATZ3 and SlPLATZ4 had variable numbers of β-strands and coils, which were 5–8 and 10–12, respectively (Appendix A).

The prediction of the binding residues based on an alignment between the template and the generated SlPLATZ models revealed that five SlPLATZ proteins (SlPLATZ1, SlPLATZ2, SlPLATZ5, SlPLATZ13, and SlPLATZ17) showed binding affinity to nucleic acids, while eight proteins (SlPLATZ3, SlPLATZ4, SlPLATZ6, SlPLATZ7, SlPLATZ9, SlPLATZ15, SlPLATZ16, and SlPLATZ19) displayed binding affinity to zinc. SlPLATZ18 was the only protein that showed an affinity to both nucleic acids and zinc. The remaining six proteins that lacked binding affinity to both nucleic acids and zinc (SlPLATZ8, SlPLATZ10, SlPLATZ11, SlPLATZ12, SlPLATZ14, and SlPLATZ20) were predicted to bind to other diverse ligands such as peptide, mannose, copper, calcium, magnesium, cyanide, phosphate, glutathione, chlorophyll-a, and glycine (Appendix A).

Gene ontology (GO) terms predicted from the modeling of SlPLATZ proteins by the I-TASSER web tool suggested that the modeled SlPLATZ proteins might function in diverse biological processes, such as cellular defense response, transcriptional regulation, gene expression, translation, protein maturation, and protein metabolic processes (Appendix A).

### 2.7. Expression Profiling of Tomato PLATZ Genes in Various Organs

To further explore the possible functional role of *SlPLATZ* genes in developmental processes, we studied their expression patterns in different tomato tissues (leaves, stems, roots, flowers, 1 cm fruits, IM fruits, MG fruits, B fruits, and B5 fruits) through RT−qPCR analysis in the tomato cultivar Ailsa Craig. The higher homology in the cDNA sequences of most *SlPLATZ* genes inhibits the availability of the sequence regions to design the primers for the expression analysis (Appendix A). Thus, nine tomato *PLATZ* genes, viz., *SlPLATZ1*, *SlPLATZ5*, *SlPLATZ10*, *SlPLATZ11*, *SlPLATZ12*, *SlPLATZ14*, *SlPLATZ17*, *SlPLATZ19*, and *SlPLATZ20*, were studied for expression profiling in various tomato organs. The expression of *SlPLATZ5*, *SlPLATZ14*, and *SlPLATZ20* was not determined in any of the organs studied. Among the remaining six *SlPLATZ* genes, two genes (*SlPLATZ10* and *SlPLATZ17*) displayed tissue-specific expression, while the other genes (i.e., *SlPLATZ1*, *SlPLATZ11*, *SlPLATZ12*, and *SlPLATZ19*) exhibited varied transcript levels across the different organs tested (Figure 6).

The transcripts of *SlPLATZ10* were detected solely in 1 cm fruits but not in any other organs examined. Likewise, the expression of *SlPLATZ17* was determined in no organ, except in flowers. *SlPlATZ1* was expressed in all tomato organs, except in breaker 5 fruits. Relative to the control (leaves), *SlPlATZ1* showed peak expression in 1 cm fruits (~12-fold), which was followed by flowers (~10-fold) and immature green fruits (~6-fold), respectively. It is noteworthy that the expression of *SlPlATZ1* was higher in the early developmental stages of tomato fruits, and then gradually diminished at the late developmental stages. The transcript levels of *SlPlATZ11* were higher (>6-fold) in flowers, roots, and stems compared with the control. However, its expression was downregulated (2.7-fold relative to the control) in 1 cm fruits and was not determined at all in the remaining developmental stages of tomato fruits.

The expression of *SlPlATZ12* was detected in various organs tested, and it showed the highest expression (>15-fold) in stems, followed by roots (>7-fold), and 1 cm fruits (2.6-fold), respectively, compared to the control. However, in contrast to the control, the transcript levels of *SlPlATZ12* were downregulated in the other developmental stages of fruits (i.e., IM fruits (>2-fold), MG fruits (>1.6-fold), breaker fruits (>3.6-fold), and breaker 5 fruits (>8.8-fold)). *SlPlATZ19* showed the highest transcription levels in flowers (>1.5-fold over the control). Its expression is also higher in the vegetative organs (i.e., leaves, stems, and roots) compared with the five fruit developmental stages. The transcripts of *SlPlATZ19* were reduced in 1 cm fruits (2.5-fold), IM fruits (2.7-fold), MG fruits (4.4-fold), breaker fruits (~2-fold) and breaker 5 fruits (4.5-fold) in comparison with the control.

### 2.8. Expression Analysis of SlPLATZ Genes in Response to Abiotic Stresses and Phytohormone Treatment

To explicate the probable functions of *SlPLATZ* genes in response to various abiotic stresses and the stress hormone ABA, the expression profiles of *SlPLATZ* genes (*SlPLATZ1*, *SlPLATZ15*, *SlPLATZ10*, *SlPLATZ11*, *SlPLATZ12*, *SlPLATZ14*, *SlPLATZ17*, *SlPLATZ19*, and *SlPLATZ20*) in the leaves sampled before and after exposure to different stress treatments were analyzed via qRT−PCR assay. *SlPLATZ5*, *SlPLATZ10*, *SlPLATZ14*, *SlPLATZ17*, and *SlPLATZ20* did not display detectable expression in all leaves harvested from control, abiotic stress-, or ABA-treated plants, while the other tomato *PLATZ* genes (*SlPLATZ1*, *SlPLATZ11*, *SlPLATZ12*, and *SlPLATZ19*) were differentially expressed under these treatments (Figure 7A–E).

*SlPLATZ1*, *SlPLATZ12*, and *SlPLATZ19* displayed responses to drought treatment, while the transcript levels of *SlPLATZ11* were not significantly altered at all time points of the stress period (Figure 7A). In comparison with the control (0 h), the transcript levels of *SlPLATZ1* were significantly induced by >2-fold at 24 h after drought treatment and peaked (>3-fold) at 48 h in response to drought stress. Although the expression of *SlPLATZ1* was unchanged at 60 h upon exposure to drought conditions, it was upregulated (>2-fold vs. the control) again at the last stage of the drought period. Similarly, *SlPLATZ19* was highly up-regulated (2.9- to 8.2-fold) at all stages of drought treatment relative to the control. By contrast, the mRNA transcript levels of *SlPLATZ12* were considerably downregulated (>2- to >55-fold relative to the control) throughout the drought stress period.

Under salt stress, the transcripts of *SlPLATZ1* were initially repressed by ~2-fold at 1 h after salt treatment compared with the control, but significantly upregulated (2.3-fold) at the last time point of the stress period. *SlPLATZ12* expression was induced (>1.6-fold over the control) at 3 h of exposure to saline conditions but decreased from 1.5- to 1.8-fold at the later stages of salt treatment. Salinity stress attenuated the expression of *SlPLATZ11* and *SlPLATZ19* from 2.4 to 3-fold at 1 h and 9 h of the stress period, respectively (Figure 7B).

In response to low-temperature conditions, *SlPLATZ11* showed upregulation from 1.6- to 2-fold relative to the control at all stages of cold treatment. Likewise, the expression levels of *SlPLATZ12* were induced by 1.6- to 2-fold throughout the cold stress period. By contrast, the expression of *SlPLATZ1* was downregulated (1.9- to 2.6-fold over the control) at all time points following exposure to cold treatment. The transcripts of *SlPLATZ19* were initially unchanged at 1 h following cold exposure but decreased significantly, 1.3- to 4-fold, at the later time points of low-temperature stress (Figure 7C).

Under heat stress, the expression of *SlPLATZ1* initially declined by ~2-fold at 1 h of exposure to high-temperature conditions but recovered at 3 h and subsequently rose (2- to 4-fold) at 9 h and 24 h compared to the control. *SlPLATZ11* was downregulated by 2.3-fold at 1 h after heat exposure but significantly upregulated (1.6- to 2-fold) at 3 h and 24 h. Similarly, the transcript accumulation levels of *SlPLATZ12* were attenuated (2-fold vs. the control) at 1 h following exposure to a high temperature but rose 2.1-fold at 3 h. On the contrary, *SlPLATZ19* was significantly repressed by 1.6- to 3.3-fold at most stages of heat treatment (Figure 7D).

The expression levels of *SlPLATZ1* were remarkably reduced (1.5- to 3.4-fold relative to the control) in the early phases of ABA treatment but significantly elevated (1.5- to 2-fold) at the later stages. *SlPLATZ12* was repressed by >1.8-fold at 1 h following exposure to ABA stress, whereas *SlPLATZ12* was minimally upregulated (1.3-fold) at 3 h. By contrast, the transcripts of *SlPLATZ19* were not significantly altered upon exposure to exogenous ABA application (Figure 7E).

### 2.9. Weighted Gene Co-Expression Network (WGCNA) Analysis and Functional Enrichment Analysis

WGCNA was conducted to construct the co-expression network of *SlPLATZ* genes using RNA-seq data (Figure 8). A total of 150 genes were detected in the co-expression networks of *SlPLATZ* genes. Specifically, 116, 10, and 24 genes were involved in the gene co-expression networks of *SlPLATZ17*, *SlPLATZ18*, and *SlPLATZ19*, respectively. The KEGG enrichment analysis of the co-expressed genes indicated that the co-expressed genes in the network of *SlPLATZ* genes were implicated in diverse biological pathways, including plant–pathogen interaction, RNA transport, mismatch repair, MAPK signaling pathway, biosynthesis of secondary metabolites, fatty acid biosynthesis, biosynthesis of amino acids, phenylpropanoid biosynthesis, nitrogen metabolism, protein export, and so forth (Figure 9, Appendix A). However, certain co-expressed genes have not been annotated in any biological process. Intriguingly, *SlSHMT2* (Solyc02g091560.3) and a *RIN--G finger* gene (Solyc08g081370) responsive to multiple abiotic stresses, *SnRK1* (Solyc02g067030) and *SlZF-31* (Solyc08g063040) associated with abiotic stress tolerance of tomato, a *NAC* gene (Solyc04g005610) responsive to drought and salinity treatments, an abiotic stress-inducible gene *SlGS2* (Solyc01g080280), *AHL5* (Solyc08g008030) implicated in the resistance to the oomycete pathogen *Phytophthora capsica*, and *Glutaredoxin* (Solyc08g062970) responsive to pathogen infection were co-expressed with the hub gene *SlPLATZ19*. *SlGS2* (Solyc01g080280) inducible by various abiotic stresses, a stress-inducible gene coding for Chaperonin 21 (Solyc12g009250), and *SnRK1* (Solyc02g067030) associated with stress tolerance in tomato were also present in the co-expression network of *SlPLATZ18*. Furthermore, *SlPLATZ17* showed co-expression with several abiotic stress-associated genes, such as *SlAPRR5* (Solyc03g081240) inducible by seawater stress, *SlTLP6* (Solyc04g071750) responsive to osmotic stress, a gene encoding nucleolar protein 6 (Solyc02g085230) inducible by aluminum stress, and low-temperature inducible genes *SlGRAS4* (Solyc01g100200) and *SlPLDβ1*(Solyc08g080130). We also identified several genes inducible by pathogen infection in the co-expression network of *SlPLATZ17*, such as *SlPP2C48* (Solyc06g076100), and genes that codify a GRAS transcription factor (Solyc07g063940), a major latex-like protein (Solyc08g023660), and a leucine-rich repeat receptor-like serine/threonine protein kinase (Solyc11g017270) (Figure 8).

### 2.10. Subcellular Location Analysis of SlPLATZ Proteins

The subcellular locations of SlPLATZ proteins predicted by the WoLF-PSORT server indicated that they were localized to the nucleus or cytoplasm (Table 1). For the further verification of their subcellular localization, full-length cDNAs of three *SlPLATZ* genes (*SlPLATZ11*, *SlPLATZ12*, and *SlPLATZ19*) were fused to the green fluorescent protein (GFP) and transiently expressed in rice protoplasts. Confocal imaging of protein fluorescence revealed that the fluorescence signals of SlPLATZ11 were observed not only in the nucleus but also in a large part of the cytoplasm. SlPLATZ12 was exclusively localized to the nucleus, while SlPLATZ19 was localized to the nucleus with its strong fluorescence signals detected in the cytosol (Figure 10).

## 3. Discussion

*PLATZ* genes, identified specifically in the plant kingdom, are novel zinc-dependent DNA-binding transcription factors that constitute a multi-gene family across a wide variety of plant species [14,32,36,37,38,39,40]. In support of previous findings, we identified 20 *PLATZ* genes in the tomato genome, revealing their existence as a multiple-gene family in tomato. This finding suggested that *PLATZ* genes might play pivotal biological roles in several diverse plant species.

Our phylogenetic analysis divided the PLATZ proteins from various plant species into two major clades (Figure 2). Of these clades, only clade I contained the PLATZ homologs from the green alga *Chlamydomonas reinhardtii*, suggesting that the proteins from clade I, especially groups 2 and 3, may be more primitive compared with those from clade II, and they might have evolved before the divergence of chlorophytes and streptophytes over one billion years ago [43]. The phylogenetic groups 4, 5, and 7 of clade II contained solely clusters of PLATZ homologous proteins from angiosperms, suggesting that the corresponding proteins from these groups might have arisen before the monocot–dicot split occurred (~200 million years ago) [44]. Intriguingly, group 2 of clade I predominantly consisted of the corresponding proteins from solanaceous crops (tomato and potato) together with the homolog from Arabidopsis (AtPLATZ6) and green alga (CrePLATZ3), suggesting that these members from Arabidopsis and solanaceous crops in group 2 may have emerged from the same progenitor before the divergence of algae and streptophytes. Notably, the tomato PLATZ proteins formed a cluster with their orthologs from potato in groups 1, 4, and 7 with 100% bootstrap support, underlying the evolutionary conservation of these PLATZ members in the solanaceous family.

Conserved domain analysis revealed that the majority of tomato PLATZ proteins were composed of only a single PLATZ domain, while an additional domain was identified in a few of them (Figure 3). This finding indicated that the integration of additional domains in SlPLATZ proteins may contribute to the diversification and enlargement of the *SlPLATZ* gene family. In addition, the similarity between the first conserved region of the PLATZ domain with certain double zinc finger domains, namely LIM (C_2_HC_5_) and RING (C_3_HC_4_) [11], and the second region of the PLATZ domain with the GATA finger (C_2_C_2_) [9] suggested that the conserved regions of the plant-specific PLATZ domain might have evolved and diversified from its ancestral zinc finger motifs through nucleotide substitution and minor insertion/deletion mutations.

Analysis of the conserved motifs and exon–intron composition in *SlPLATZ* genes revealed the structural similarity among the majority of *SlPLATZ* genes clustered in the same phylogenetic groups, suggesting their structural conservation in the more closely related *SlPLATZ* family members (Appendix A). The similar motif composition of paralogous and orthologous gene pairs from tomato, Arabidopsis, and rice suggested the important role of these conserved motifs in the expansion of the *PLATZ* gene family and the evolutionary conservation of those motifs during species evolution.

Gene duplication is an important evolutionary mechanism that enhances the adaption of organisms to diverse environments by generating additional genes [45,46]. The prediction of three segmentally duplicated gene pairs and four tandemly duplicated gene pairs in the *SlPLATZ* gene family suggested that both segmental and tandem gene duplication triggered the expansion of the tomato *PLATZ* gene family during evolution. Our finding is in line with the previous reports that identified several duplicated gene pairs responsible for the enlargement of the *PLATZ* gene family in multiple plant species [14,32,38,39,40]. Group-specific gene duplication events were observed in the tomato *PLATZ* gene family since the duplicated genes of each pair belonged to the same phylogenetic groups (Table 2).

The microsynteny analysis revealed more orthologous gene pairs between tomato and Arabidopsis and very few orthologs between tomato and rice (Figure 4). This result showed that tomato is evolutionarily more closely related to the dicotyledonous model plant Arabidopsis, in comparison with the monocotyledonous model plant rice. Two *SlPLATZ* genes were collinear with those of both Arabidopsis and rice, suggesting that these orthologous genes in tomato, Arabidopsis, and rice may have existed before the divergence of the ancestral genes during the evolutionary process.

The upstream regions of numerous stress-related genes have been identified to contain a variety of stress-associated *cis*-regulatory elements [47,48,49]. The prediction of a diverse number of stress-responsive and hormone-related *cis*-acting elements in the promoter regions of many *SlPLATZ* genes (Appendix A and Appendix A) supported the abiotic stress responsiveness of *SlPLATZ* genes. For instance, the presence of the ABA-responsive element ABRE and the low-temperature responsive element LTR in the promoter region of *SlPLATZ1*, the LTR element and the drought-responsive element MBS in that of *SlPLATZ12*, and the LTR element in that of *SlPLATZ19* suggested that these *cis* elements might act as molecular switches in their response to these stress conditions. Our result was corroborated by the presence of stress-responsive *cis*-elements in the promoters of *PLATZ* family members in diverse plant species [38,39,40].

Furthermore, fifteen tomato miRNAs implicated in the development and stress tolerance of tomato (namely, *sly-miR159*, *sly-miR162*, *sly-miR171a*, *sly-miR319a*, *sly-miR319b*, *sly-miR319c-3p*, *sly-miR396a-5p*, *sly-miR396b*, *sly-miR403-5p*, *sly-miR482a*, *sly-miR5303*, *sly-miR6024*, *sly-miR9469-3p*, *sly-miR9472-3p*, and *sly-miR9478-3p*) targeted to cleave twelve *SlPLATZ* genes [50,51,52,53,54,55,56,57,58,59], hinting at the miRNA-mediated regulation of SlPLATZ transcription factors in the development and stress adaptation of tomato (Appendix A). For example, *sly-miR396a-5p* and *sly-miR396b* are involved in drought stress response in tomato [53,54]. The presence of their target site in *SlPLATZ19* suggests they might regulate *SlPLATZ19* in the drought acclimation of tomato. Interestingly, *SlPLATZ11*, *SlPLATZ13,* and *SlPLATZ17* were predicted to be the targets of multiple miRNAs. These findings suggested that multiple miRNAs may regulate the expression of a single *SlPLATZ* gene in the stress adaptation of tomato.

To further explore the molecular structure and functions of SlPLATZ proteins, a 3D structure prediction was conducted (Figure 5). The comparative modeling of SlPLATZ proteins predicted their binding affinities with diverse ligands, including nucleic acids and peptide molecules. Furthermore, gene ontology (GO) annotations for the obtained SlPLATZ models suggested their putative functions in a variety of biological processes, including cellular defense response, transcriptional regulation, and protein metabolic processes (Appendix A). This result suggested their crucial biological roles in tomato, including functioning as transcription factors.

Differential expression of genes in various tissues provides valuable clues about their functional diversity and possible roles in developmental processes. The tissue-specific expression and differential expression patterns of tomato *PLATZ* gene family members in different organs suggested their functional diversity in the development of tomato (Figure 6). This result is consistent with the tissue-specific expression and varied expression in various organs of *PLATZ* genes in other plant species [36,39].

The mRNA transcripts of *SlPLATZ5*, *SlPLATZ14*, and *SlPLATZ20* were undetectable in any organ examined, suggesting that they could be pseudogenes in the tomato genome or might be expressed only at certain critical developmental stages. In addition to providing mechanical strength to the aerial parts of plants, the stem mediates the long-distance movement of water and minerals to facilitate plant growth under both favorable and unfavorable conditions. The predominant expression of *SlPLATZ12* in stems hinted at its putative function in the development of stems and the abiotic stress adaptation of tomato. Flowering and floral development, regulated by numerous floral genes and various environmental factors, are prerequisites for fruit set and important for crop yield [60]. The flower-specific expression of *SlPLATZ17* indicated its potential function in the development of flower organs in tomato. Besides *SlPLATZ17*, the transcript levels of *SlPLATZ11* and *SlPLATZ19* were the most abundant in flowers, revealing their likely function in the development of flowers (Figure 6).

*SlPLATZ10* is a fruit-specific gene that was expressed preferentially in 1 cm fruits but not in any other organs. This finding suggested that it may have a specific function in the initial developmental phase in tomato. *SlPLATZ1* showed the highest expression in 1 cm fruit, but its expression gradually declined at later fruit developmental phases, highlighting its putative function in different developmental stages of tomato fruit, especially in the cell division stage of tomato (Figure 6).

Plant responses and adaptation to a wide range of abiotic stimuli are closely linked to differential gene expression mediated by a complex network of multiple transcription factors and a variety of stress-related genes via an ABA-dependent or ABA-independent pathway [42,61,62]. The functions of PLATZ transcription factors in plant abiotic stress tolerance and their responses to various stress and hormone treatments have been well documented [14,32,40]. Here we showed that tomato *SlPLATZ* genes (i.e., *SlPLATZ1*, *SlPLATZ11, SlPLATZ12*, and *SlPLATZ19*) were also responsive to different abiotic stress treatments (Figure 7A–E), suggesting their functional role in abiotic stress responses of tomato. Drought stress triggered significant alterations in the expression levels of *SlPLATZ1, SlPLATZ12*, and *SlPLATZ19*. The transcripts of *SlPLATZ1* and *SlPLATZ19* were markedly induced at most or all stages of drought treatment, respectively, whereas the expression of *SlPLATZ12* was considerably suppressed at all time points following exposure to drought (Figure 7A). Our finding on the responses of *SlPLATZ* genes to drought stress is well correlated with a previous report in which the overexpression of *PLATZ* genes from bamboo and Arabidopsis was observed to promote drought tolerance in the transgenic Arabidopsis lines with respect to the wild-type plants [23,24,25], and another report in which the expression of *PLATZ-TF7* (Zm00001d051511) was induced in drought-tolerant maize plants under drought treatment [31].

On exposure to salinity stress, *SlPLATZ* genes (*SlPLATZ1*, *SlPLATZ11, SlPLATZ12*, and *SlPLATZ19*) displayed remarkable changes in their transcription levels (Figure 7B). The expression of *SlPLATZ12* and *SlPLATZ19* was up- or down-regulated under salt treatment, whereas that of *SlPLATZ11* and *SlPLATZ19* decreased (2- to 3-fold over the control) in the early phase and late phase of the saline stress period, respectively. These responses of *SlPLATZ* genes under salt treatment revealed their probable involvement in the salt stress adaptation of tomato. This result is consistent with a previous report wherein *GhPLATZ1*-overexpressing transgenic Arabidopsis plants showed robust resistance to osmotic and salt stress [28].

In response to low temperature, the expression of *SlPLATZ* genes was significantly up- or down-regulated. The transcription levels of *SlPLATZ11* and *SlPLATZ12* were sharply elevated throughout the cold stress period. Conversely, *SlPLATZ1* and *SlPLATZ19* were significantly downregulated following cold exposure (Figure 7C). These findings highlighted the fact that tomato *PLATZ* genes may also play a role in the adaptation of tomato to low-temperature stress. Our finding is supported by recent work reporting that the expression of most *PhePLATZ* genes is induced in response to cold treatment [32].

High-temperature stress also remarkably altered the expression profiles of *SlPLATZ* genes (Figure 7D). *SlPLATZ1, SlPLATZ11*, and *SlPLATZ12* exhibited similar responses to heat stress, with significantly downregulated transcript levels at the early stages of heat treatment, followed by sharply upregulated expression levels at the later stages. *SlPLATZ19* was markedly downregulated in multiple phases of the stress period, indicating the putative functions of *SlPLATZ* genes in the heat stress response of tomato. This result is corroborated by a previous study reporting the upregulation of the *PLATZ* gene (Ca01g07220) in the heat-tolerant pepper cultivar compared with the sensitive cultivar [30].

The regulation of various stress-associated genes by the stress hormone ABA has been extensively investigated in order to enhance plant adaptation to unfavorable environmental stresses, including drought, salt, cold, and heat stress [63,64,65]. *SlPLATZ1* showed remarkable responses to ABA treatment, while the expression of *SlPLATZ19* was not significantly altered and that of *SlPLATZ11* and *SlPLATZ12* was responsive only at one time point following exposure to ABA treatment (Figure 7E). This finding implies that *SlPLATZ1* may have a possible function in the abiotic stress adaptation of tomato through an ABA-dependent pathway, whereas the function of the other genes (*SlPLATZ11, SlPLATZ12*, and *SlPLATZ19*) is unlikely to be related to ABA.

To further explore the functions of SlPLATZ proteins, their subcellular localization was empirically observed under a confocal microscope after the transient expression of the SlPLATZ–GFP fusion proteins in the rice protoplast. The nuclear-specific localization of SlPLATZ12 suggested its specific role in transcriptional regulation as a transcription factor. The localization of SlPLATZ11 and SlPLATZ19 in both the nucleus and the cytoplasm spotlighted the possibility that they might shuttle between the cytoplasm and the nucleus to play a regulatory role in various cellular signaling pathways in addition to their role in gene regulation as transcription factors. Our finding is consistent with the previous studies that reported the localization of PLATZ proteins in the nucleus or the nucleus and cytoplasm [26,39].

To gain a better understanding of the putative functions of *SlPLATZ* genes, the gene co-expression network analysis of *SlPLATZ*s was performed using the RNA-seq data. The co-expressed genes identified in the networks of *SlPLATZ17*, *SlPLATZ18*, and *SlPLATZ19* were implicated in various biological pathways, including stress adaptation in tomato, underlying their biological importance in tomato (Figure 8 and Figure 9, Appendix A). Expression profiling via qRT−PCR analysis revealed that *SlPLATZ19* was responsive to various abiotic stresses. This finding was corroborated by the presence of numerous stress-associated genes, such as *SlSHMT2* (Solyc02g091560.3), *SnRK1* (Solyc02g067030), *SlGS2* (Solyc01g080280), *AHL5* (Solyc08g008030), *SlZF-31* (Solyc08g063040), *Glutaredoxin* (Solyc08g062970), a *RING Finger* gene (Solyc08g081370), and a *NAC* gene, in the gene co-expression network of *SlPLATZ19* [66,67,68,69,70,71,72,73].

In addition, multiple stress-related genes, including *SlGS2* (Solyc01g080280), *SnRK1* (Solyc02g067030), and *Chaperonin 21* (Solyc12g009250), were co-expressed with *SlPLATZ18* [68,70,74]. The hub gene *SlPLATZ17* also constituted a co-expression network with several stress-responsive genes in tomato, namely *SlAPRR5* (Solyc03g081240), *SlTLP6* (Solyc04g071750), *SlGRAS4* (Solyc01g100200), *SlPLDβ1* (Solyc08g080130), *SlPP2C48* (Solyc06g076100), a gene encoding nucleolar protein 6 (Solyc02g085230), a GRAS transcription factor (Solyc07g063940), a major latex-like protein (Solyc08g023660), and a leucine-rich repeat receptor-like serine/threonine-protein kinase (Solyc11g017270) [75,76,77,78,79,80,81,82,83]. These results suggested that *SlPLATZ* genes may have important functions in abiotic stress tolerance in tomato.

## 4. Materials and Methods

### 4.1. Identification and Sequence Analysis of PLATZ Genes in the Tomato Genome

We downloaded *Arabidopsis thaliana* PLATZ protein sequences from TAIR https://www.Arabidopsis.org/ (accessed on 31 March 2022) and the HMM profile of SlPLATZ (PF04640) from the Pfam database http://pfam.xfam.org/ (accessed on 31 March 2022). Then, we performed basic local alignment search tool (BLAST) searches with default parameters of the Sol genomics database using AtPLATZ protein sequences and the HMM profile of SlPLATZ (PF04640) as queries. The resulting 20 non-redundant SlPLATZ protein sequences were validated using the SMART web tool http://smart.emblheidelberg.de/ (accessed on 1 April 2022), the NCBI CDD search https://www.ncbi.nlm.nih.gov/Structure/bwrpsb/bwrpsb.cgi (accessed on 1 April 2022), and the Pfam database to reconfirm the existence of the PLATZ domain. The protein length, molecular weight, GRAVY values (grand average of hydropathicity index), and isoelectric points of the identified SlPLATZ proteins were investigated using the Expasy server http://cn.expasy.org/tools/protparam.html (accessed on 4 April 2022). The Open Reading Frame Finder tool https://www.ncbi.nlm.nih.gov/orffinder/ (accessed on 4 April 2022) was employed to determine the open reading frames of the *SlPLATZ* genes. Clustal Omega and ESPript web tools were used for multiple-protein sequence alignment [84,85]. The Multiple EM for Motif Elicitation (MEME) web server http://meme-suite.org/ (accessed on 18 April 2022) was employed to analyze the conserved motifs in the full-length protein sequences from Arabidopsis, tomato, and rice. The analysis conditions were set as follows: a maximum number of 10 motifs and a motif length between 6 and 50 amino acids. The exon–intron structure of *SlPLATZ* genes was investigated using the Gene Structure Display Server (GSDS) http://gsds.cbi.pku.edu.cn/ (accessed on 7 April 2022). The web tool “WoLF-PSORT” https://wolfpsort.hgc.jp/ (accessed on 6 May 2022) was used to predict the subcellular localization of the identified SlPLATZ proteins.

### 4.2. Phylogenetic Analysis of Tomato PLATZ Proteins

The full-length PLATZ protein sequences from 10 plant species were aligned using ClustalW, which was followed by phylogenetic analysis using the neighbor-joining (NJ) method with 1000 bootstrap replications in MEGA 6.0 [86]. The deduced PLATZ protein sequences of *Chalamdomonas renhadi*, *Physcomitrium patens*, *Amborella trichopoda*, tomato, potato, Arabidopsis, and *Sorghum bicolor* were retrieved from the Phytozome database http://www.phytozome.net (accessed on 31 March 2022), while those of rice, maize, and *Brassica rapa* were obtained from the literature [38]. The gene names and accession numbers used in the phylogenetic tree are shown in Appendix A.

### 4.3. Chromosomal Locations, Gene Duplication, and Microsynteny Analysis

The chromosomal locations of *SlPLATZ* genes were investigated using the Sol genomic database and then visualized with TBtools. Gene duplications among *SlPLATZ* genes were analyzed with the one-step MCScanX program of TBtools software [87] and examined by BLASTP with an E-value < 10^−10^. The synonymous (Ks) and non-synonymous (Ka) nucleotide substitution rates of duplicated *SlPLATZ* gene pairs were investigated using the method of Nei and Gojobori [88]. The mode of selection was determined by the analyzed Ka/Ks ratio [89]. The divergence time (T) of the predicted duplicated gene pairs was estimated using the formula T = Ks/2r MYA (millions of years ago). Ks is the synonymous substitution rate per site, and r is considered the constant for dicotyledonous plants of 1.5 × 10^−8^ substitutions per site per year [90]. The microsynteny analysis of *PLATZ* genes across tomato, Arabidopsis, and rice was conducted using a reciprocal BLAST search approach against the entire genomes of these species. The duplicated gene pairs were visualized using TBtools software [87].

### 4.4. Prediction of Cis-Acting Elements and miRNA Target Sites

The psRNATarget online tool http://plantgrn.noble.org/psRNATarget/analysis (accessed on 16 July 2022) was employed to predict the putative miRNA sequences in tomato *PLATZ* genes. The putative *cis*-regulatory elements located within 1500 bp upstream from the transcription start point of *SlPLATZ* genes were analyzed using the PlantCare database http://bioinformatics.psb.ugent.be/webtools/plantcare/html/ (accessed on 11 April 2022).

### 4.5. Homology Modeling of SlPLATZ Proteins

The protein sequences of SlPLATZ1-20 were used as input to perform the 3D structure prediction through the I-TASSER server. The 3D models of tomato PLATZ proteins were generated from multiple threading alignments with LOMETS and iterative TASSER assembly simulations [91]. The template analogs were identified, and the best-modeled structures were selected based on the maximum scores. The generated 3D models were refined using ModRefiner [92]. The final modeled 3D structures and ligand-binding sites of SlPLATZ proteins were visualized with Discovery Studio v.21.1. The putative functions of modeled PLATZ proteins were predicted via the I-TASSER server according to global and local similarity to template proteins curated in the PDB with known structures and functions.

### 4.6. Plant Sample Collection and Stress Treatments

Tomato seeds (*Solanum lycopersicum* L. cv. Ailsa Craig) were germinated in commercial horticultural soil for seedlings and maintained in a growth chamber controlled at 25 °C day/20 °C night, a 16-h light/8-h dark photoperiod, a relative humidity of 55–70%, and a light intensity of 300 μmol m^−2^ s^−1^ until they developed into 28-day-old seedlings. Following the collection of fresh roots, stems, and leaves from 28-day-old plants for organ-specific expression analysis, the remaining plants were moved to a greenhouse adjusted at 25/20 °C day/night temperatures until they reached the reproductive stage for the harvesting of flower and fruit samples. For the expression profiling at different fruit developmental stages, the following samples were collected: (i) young fruits approximately 7 days after pollination and 1 cm in diameter (1 cm fruits), (ii) immature fruits approximately 21 days after pollination (IM fruits), (iii) mature green fruits approximately 35 days after pollination (MG fruits), (iv) fruits at the breaker stage when the green color of mature fruits alters to light yellow-orange (B fruits), and (v) fruits 5 days after the breaker stage (B5 fruits) [49].

Leaf samples from 28-day-old tomato seedlings treated with various abiotic stress stimuli (salt (NaCl), drought, heat, cold, and abscisic acid (ABA)) were collected to analyze the expression patterns of *SlPLATZ* genes in response to different abiotic stress conditions. The transcript levels of *SlPLATZ* genes were monitored at 0, 1, 3, 9, and 24 h after treatment with heat, cold, salt (NaCl), and abscisic acid (ABA), and at 0, 24, 48, 60, and 72 h after drought treatment. Drought stress was imposed on the tomato plants by withholding water for 72 h [93]. ABA treatment was applied to the plants by spraying leaves with 100 μM ABA [94]. To impose heat and cold stress, the tomato seedlings were incubated in a growth chamber adjusted at 40 °C and 4 °C, respectively. Salt stress was induced by immersing roots in a nutrient solution with 200 mM NaCl, and the plants in the nutrient solution without salt were used as the 0 h control. Tomato seedlings grown in soil under normal conditions (25 °C) served as the 0 h controls for heat, cold, drought, and ABA treatment [95]. All plant samples were collected from three biological replicates, immediately frozen in liquid nitrogen, and stored at −80 °C for further use.

### 4.7. Expression Profiling of SlPLATZ Genes by qRT−PCR

Total RNA was extracted from all samples using an RNeasy Mini kit (Qiagen, Hilden, Germany) and purified with an RNase-free DNase I kit (Qiagen, Hilden, Germany) following the manufacturer’s instructions. The quality and quantity of extracted RNA were measured using a NanoDrop^®^ 1000 spectrophotometer (Wilmington, DE, USA). A Superscript^®^ III First-Strand cDNA synthesis kit (Invitrogen, Carlsbad, CA, USA) was used to synthesize cDNA from 1μg of total RNA in accordance with the manufacturer’s protocols. The gene-specific primers for *SlPLATZ* genes were designed using Primer3 software http://frodo.wi.mit.edu/primer3/input.htm (accessed on 25 April 2022) (Appendix A), and a melting curve analysis was conducted to validate the specificity of each primer set [95]. The *18S rRNA* (F: AAAAGGTCGACGCGGGCT, R: CGACAGAAGGGACGAGAC) from tomato was used as a control gene for normalization [96]. qRT-PCR analysis was conducted in a total volume 10 μL reaction mixture comprising 1 μL (50 ng) cDNA, 2 μL forward and reverse primers of 5 pmol concentration, 5 μL of iTaq SYBR Green (Qiagen, Hilden, Germany), and 2 μL double distilled water. A LightCycler^®^ 96SW 1.1 (Roche, Germany) was used for amplification and determination of the Cq value of the samples with the following PCR parameters: pre-denaturation at 94 °C for 5 min, followed by 40 cycles at 94 °C for 15 s, annealing at 60 °C for 20 s, and extension at 72 °C for 30 s. The 2^−∆∆Ct^ method was employed to analyze the relative expression of each gene against each treatment [97].

### 4.8. Co-Expression Network Analysis of SlPLATZ Genes

Raw RNA-Seq data used for the gene co-expression network analysis were downloaded from the NCBI SRA database (Appendix A). The quality assessment on raw sequence reads was performed using the FastQC toolkit [98]. The raw reads were cleaned by removing low-quality reads and adapter contamination. HISAT2 2.1.0 was used to map the cleaned reads against the tomato reference genome ITAG4.0 [99]. The number of reads that mapped on exons was analyzed with FeatureCounts 1.6.2 [100]. The expression was analyzed as the fragment per kb per million reads (FPKM) by using DESeq2 software [101]; removeBatchEffect processing was carried out to remove the batch effect in R [102]. For the construction of the co-expression network of *SlPLATZ* genes, weighted gene co-expression network analysis (WGCNA) was conducted using those genes with an FPKM greater than 1 in R [103]. The co-expression networks were constructed using Cytoscape https://cytoscape.org/ (accessed on 29 June 2022). We used the top co-expressed genes from the network figures for GO and KEGG analysis, which was accomplished using the KOBAS web tool http://kobas.cbi.pku.edu.cn/ (accessed on 11 July 2022).

### 4.9. Subcellular Localization

The coding regions of *SlPLATZ* genes were amplified with gene-specific primers (Appendix A) and then cloned in the pGA3452 vector driven by the maize Ubi1 promoter to generate SlPATZ –GFP fusion proteins [104]. A vector that expresses an NLS-mRFP fusion protein served as a nuclear marker. The SlPLATZ–GFP fusion construct and the NLS-mRFP construct were introduced into rice Oc cell protoplasts using the electroporation method [105]. After incubation overnight at 28 °C in the dark for 12 to 16 h, the resulting transformants were examined for the detection of fluorescent signals under a confocal fluorescence microscope (BX61; Olympus, Tokyo, Japan) using bright field illumination, the GFP channel, and the RFP channel.

### 4.10. Statistical Analysis

The data were analyzed using two-tailed Student’s t-tests in SigmaPlot 14 (SYSTAT and MYSTAT Products, United States and Canada). Asterisks (*, **, and ***) were used to denote the statistical significance at *p*-value <0.05, <0.01, and <0.001, respectively.

## 5. Conclusions

We identified 20 non-redundant *SlPLATZ* genes, most of which harbored two conserved regions essential for zinc-dependent DNA binding, in the tomato genome. Gene duplication analysis spotlighted the fact that both segmental and tandem gene duplication events are accountable for the expansion of the *PLATZ* gene family in tomato over the course of evolution. The potential functions of *SlPLATZ* genes in the abiotic stress acclimation of tomato were highlighted by expression profiling that revealed the responses of *SlPLATZ1*, *SlPLATZ11*, *SlPLATZ12*, and *SlPLATZ19* to diverse abiotic stresses and gene co-expression network analysis that uncovered the co-expression of several stress-related genes in the networks of *SlPLATZ11*, *SlPLATZ17*, and *SlPLATZ19*. The tissue-specific expression of *SlPLATZ10* and *SlPLATZ17*, and the varied expression levels of *SlPLATZ1*, *SlPLATZ11*, *SlPLATZ12*, and *SlPLATZ19* in diverse organs suggested their different functional roles in the development of tomato. The predominant localization of selected SlPLATZ–GFP fusion proteins in the nucleus revealed their putative function as transcription factors. Our results provide valuable information for the further functional elucidation of potential PLATZ transcription factors implicated in the development and abiotic stress tolerance of tomato.

## Figures and Tables

**Figure 1 plants-11-03112-f001:**
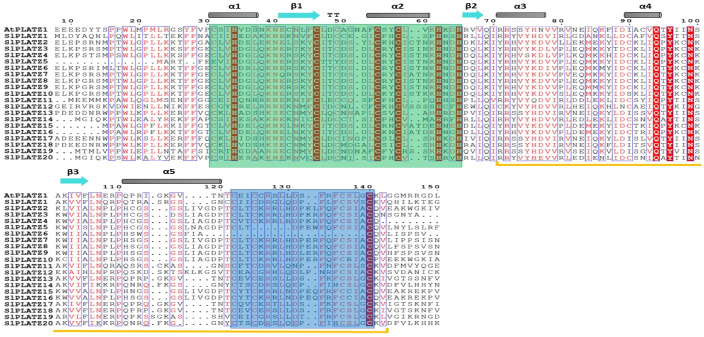
Multiple sequence alignment of all deduced tomato PLATZ polypeptides with that of Arabidopsis. The green box highlights the first conserved region [C-x_2_-H-x_11_-C-_x2_-C-x_(4–5)_-C-x_2_-C-x_(3–5)_-H-x_2_-H] and the blue box marks the second conserved region [C-x_2_-C-x_(10–12)_-C-x_3_-C], which are essential for zinc-dependent DNA binding. The yellow underline indicates the location of the PLATZ domain. The elements of the secondary structure determined using the ESPript 3.0 web server are represented by gray bars (α-helices) and blue-green arrows (β-strands) above the alignment.

**Figure 2 plants-11-03112-f002:**
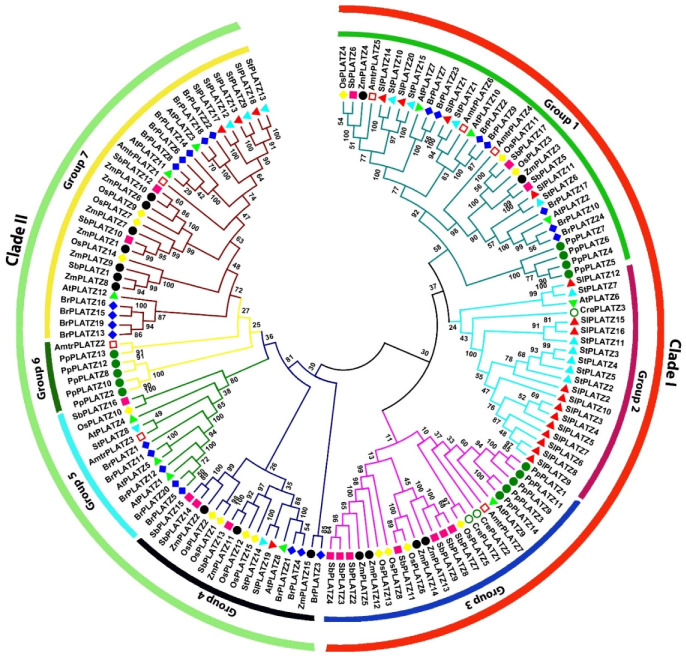
Phylogenetic analysis of PLATZ proteins. Full-length polypeptide sequences from tomato (*Solanum lycopersicum*), potato (*Solanum tuberosum*), Arabidopsis (*Arabidopsis thaliana*), rice (*Oryza sativa*), great millet (*Sorghum bicolor*), maize (*Zea mays*), Amborella (*Amborella trichopoda*), moss (*Physcomitrella patens*), and green alga (*Chlamydomonas reinhardtii*) were used to construct the phylogenetic tree by the neighbor-joining method in MEGA6 software with 1000 bootstrap replicates.

**Figure 3 plants-11-03112-f003:**
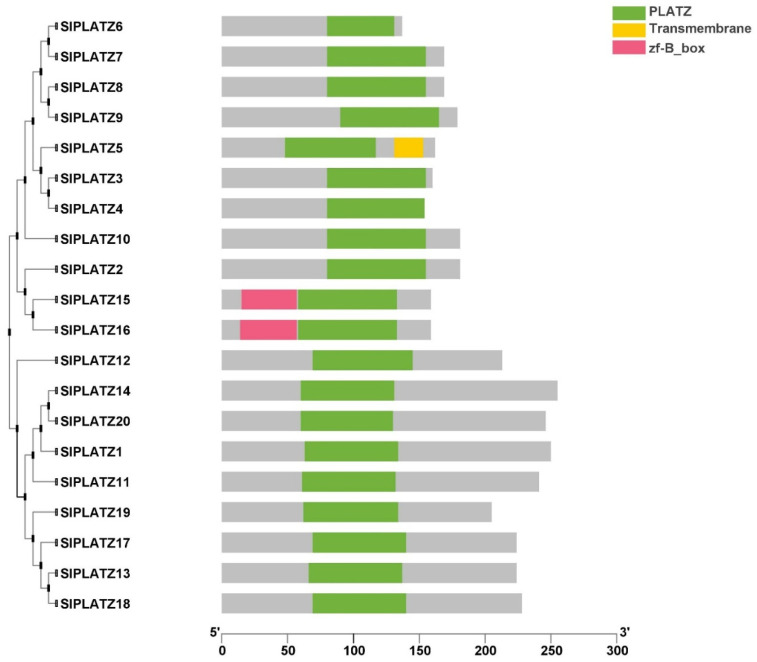
Schematic depiction of the conserved domains of the tomato PLATZ proteins. The transmembrane domain (TM), the PLATZ domain, and the B-box-type (B_box) zinc finger domain identified in SlPLATZ proteins are displayed in yellow, green, and red boxes, respectively.

**Figure 4 plants-11-03112-f004:**
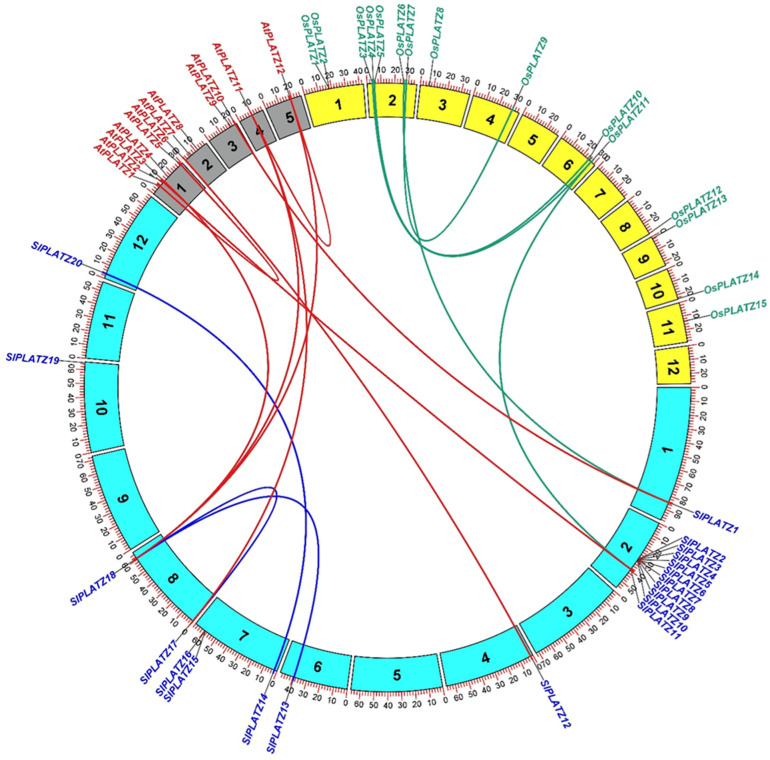
Schematic diagram depicting the microsyntenic relationship of *PLATZ* genes across Arabidopsis, tomato, and rice. The chromosomes from three plant species are illustrated with the scale in megabase pairs (Mbp) and marked in gray for Arabidopsis, yellow for rice, and aqua for tomato, respectively. Blue lines link the segmentally duplicated genes in tomato.

**Figure 5 plants-11-03112-f005:**
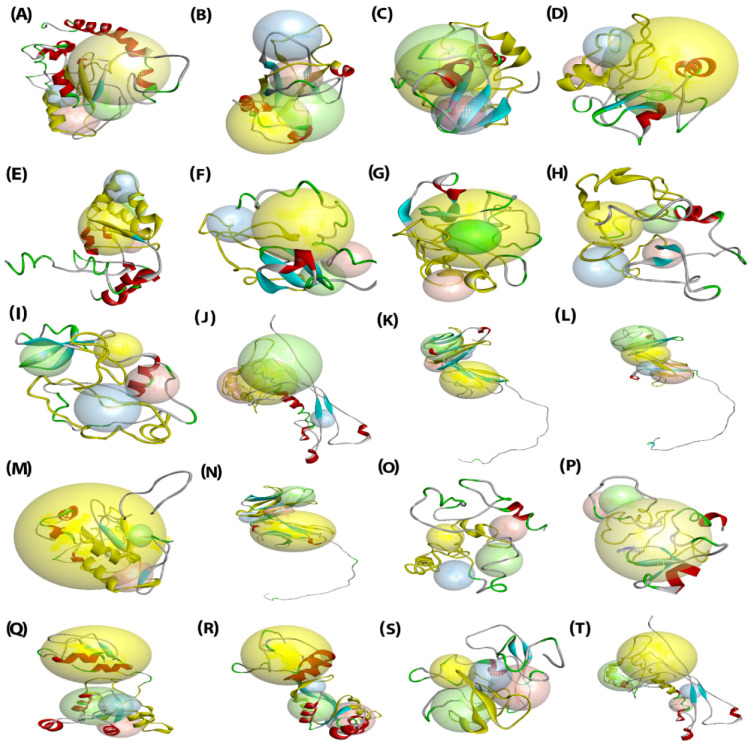
The final 3D modeled structures of SlPLATZ proteins generated using Discovery Studio v.21.1. The components of secondary structure: α-helices (red), β-sheets (cyan), coils (green), and loops (gray) and the four binding sites: site 1 (yellow sphere), site 2 (green sphere), site 3 (red sphere), and site 4 (blue sphere), are displayed in the predicted 3D structures of (**A**–**T**) SlPLATZ1–SlPLATZ20.

**Figure 6 plants-11-03112-f006:**
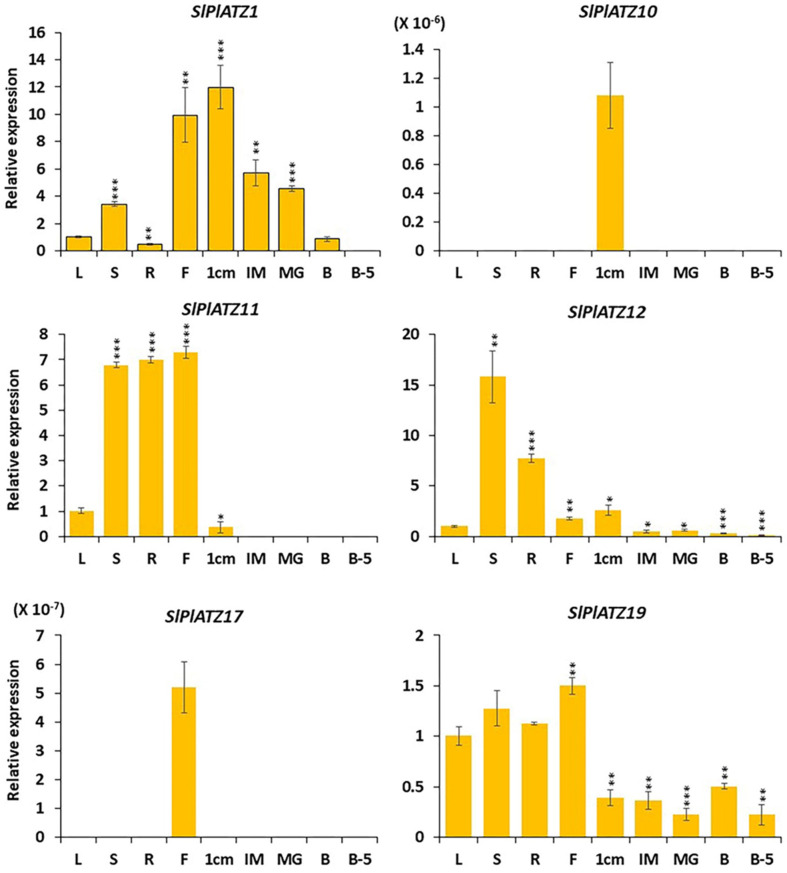
qRT−PCR expression analysis of *SlPLATZ* genes in various organs. The organs used for the expression analysis are leaves (L), roots (R), stems (S), flowers (F), 1 cm fruits (1 cm), immature fruits (IM), mature green fruits (MG), breaker fruits (B), and fruits 5 days after breaker stage (B5). Error bars indicate the standard deviations of the means of three independent biological replicates. *, **, and *** represent the significant difference between the control samples (leaves) and the samples from the other organs according to Student’s t−test, at *p*−values <0.05, <0.01, and <0.001, respectively.

**Figure 7 plants-11-03112-f007:**
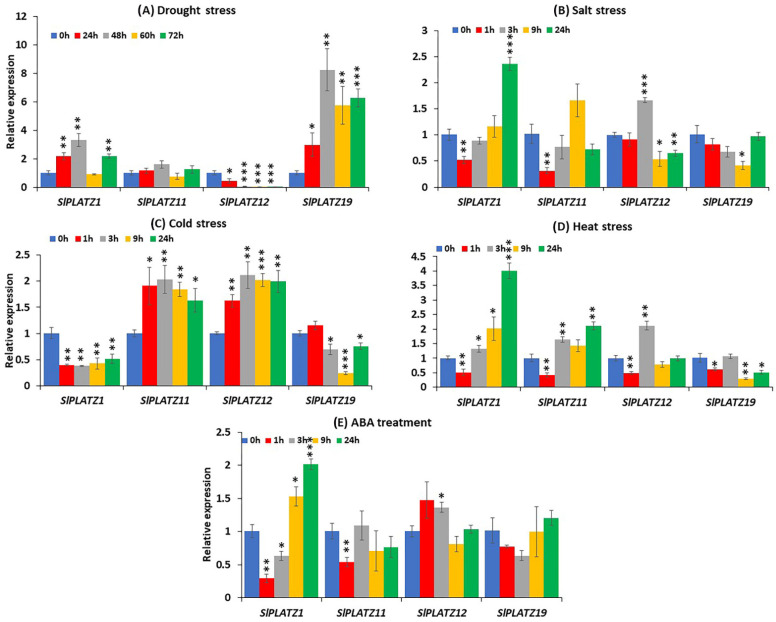
(**A**–**E**) qRT−PCR analysis of *SlPLATZ* genes on exposure to different abiotic stresses and ABA treatment. The standard deviations of the average of three biological replicates of the qRT−PCR analysis are represented by error bars. The significant difference between the control samples (0 h) and the stress-treated samples of *SlPLATZ* genes is indicated by the asterisks (*, **, and ***) determined by Student’s t−test at *p*−values < 0.05, <0.01, and <0.001, respectively.

**Figure 8 plants-11-03112-f008:**
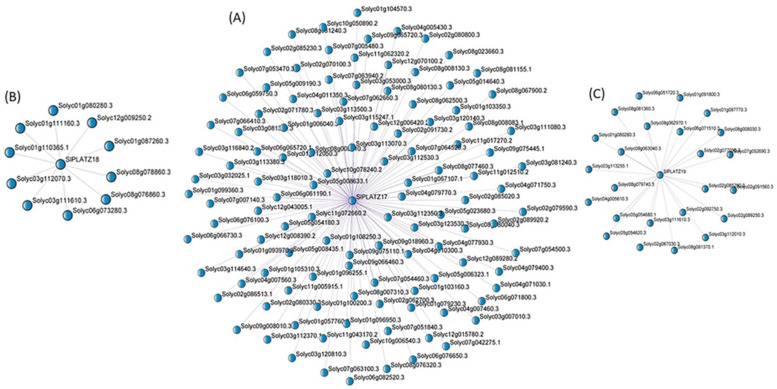
Gene co-expression network analysis of *SlPLATZ* genes. The co-expressed genes identified in the networks of *SlPLATZ17* (**A**), *SlPLATZ*18 (**B**), and *SlPLATZ19* (**C**).

**Figure 9 plants-11-03112-f009:**
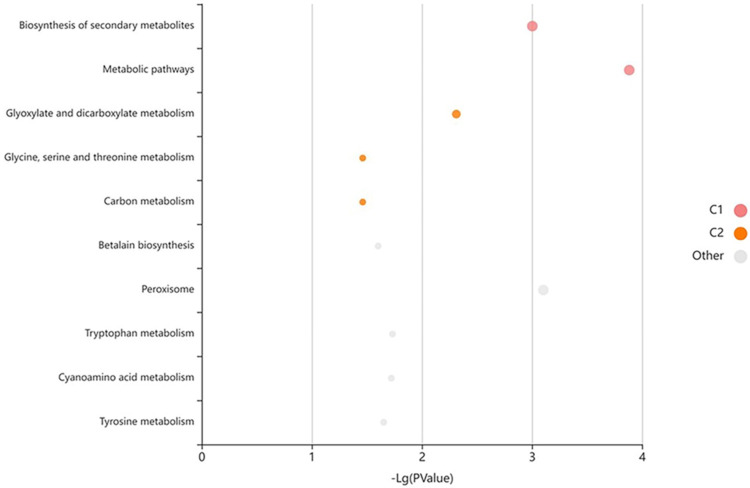
Bubble chart of the KOBAS enrichment analysis for co-expressed genes with *SlPLATZ* genes. Each bubble denotes an enriched function, and the sizes of the bubble from small to large reflect the following values: [0.05, 1], [0.01, 0.05], [0.001, 0.01], [0.0001, 0.001], [1 × 10^−10^, 0.0001], [0, 1×10^−10^].

**Figure 10 plants-11-03112-f010:**
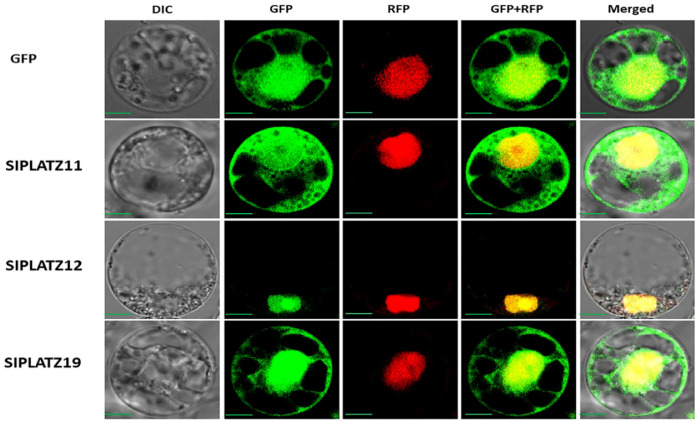
Subcellular localization analysis of SlPLATZ11, SlPLATZ12, and SlPLATZ19 by transiently expressing the SlPLATZs–GFP fusion constructs in Oc cell protoplasts. The NLS-mRFP construct served as a control for a nuclear protein. Scale bars = 100 μm.

**Table 1 plants-11-03112-t001:** In silico analysis of *PLATZ* genes and their corresponding proteins in tomato.

Gene Name	Locus Name	ORF (bp)	Chromosome No.			Protein			Subcellular Localization	No. of Introns
				Length (aa)	PLATZ Domain Start-End (aa)	MW (kDa)	pI	GRAVY		
*SlPLATZ1*	Solyc01g091000	753	1	250	63–134	27.76	8.76	−0.272	Nucleus	3
*SlPLATZ2*	Solyc02g033100	546	2	181	80–155	20.88	9.2	−0.528	Nucleus	3
*SlPLATZ3*	Solyc02g033110	483	2	160	80–155	18.41	8.69	−0.628	Nucleus	3
*SlPLATZ4*	Solyc02g033120	462	2	154	80–154	17.54	8.92	−0.568	Nucleus	2
*SlPLATZ5*	Solyc02g036120	489	2	162	48–117	19.19	8.26	−0.217	Cytoplasm	1
*SlPLATZ6*	Solyc02g036130	414	2	137	80–131	15.94	8.61	−0.382	Nucleus	2
*SlPLATZ7*	Solyc02g036140	510	2	169	80–155	19.29	8.7	−0.359	Nucleus	2
*SlPLATZ8*	Solyc02g036170	510	2	169	80–155	19.34	8.95	−0.397	Nucleus	2
*SlPLATZ9*	Solyc02g036200	540	2	179	90–165	20.49	8.86	−0.335	Nucleus	3
*SlPLATZ10*	Solyc02g036230	546	2	181	80–155	20.86	9.48	−0.709	Cytoplasm	3
*SlPLATZ11*	Solyc02g068510	726	2	241	61–132	27.39	6.64	−0.445	Nucleus	3
*SlPLATZ12*	Solyc04g008090	642	4	213	69–145	24.73	9.18	−0.57	Nucleus	2
*SlPLATZ13*	Solyc06g061240	675	6	224	66–137	25.34	9.54	−0.651	Nucleus	2
*SlPLATZ14*	Solyc07g007320	768	7	255	60–131	29.16	8.55	−0.433	Nucleus	2
*SlPLATZ15*	Solyc07g049120	480	7	159	58–133	18.58	8.89	−0.61	Cytoplasm	2
*SlPLATZ16*	Solyc07g049130	480	7	159	58–133	18.58	9.22	−0.633	Nucleus	2
*SlPLATZ17*	Solyc08g005100	675	8	224	69–140	25.62	9.37	−0.498	Nucleus	2
*SlPLATZ18*	Solyc08g076860	687	8	228	69–140	25.57	9.39	−0.393	Nucleus	2
*SlPLATZ19*	Solyc10g085800	618	10	205	62–134	22.9	9.08	−0.393	Nucleus	2
*SlPLATZ20*	Solyc12g010470	741	12	246	60–130	28.02	8.45	−0.427	Nucleus	2

ORF, open reading frame; bp, base pair; aa, amino acid; kDa, kilo Dalton; pI, isoelectric point; MW, molecular weight; GRAVY, Grand average of hydropathicity.

**Table 2 plants-11-03112-t002:** Predicted Ka/Ks ratios of the duplicated *SlPLATZ* gene pairs along with their divergence time.

Duplicated Gene Pair	Ka	Ks	Ka/Ks	Duplication Type	Types of Selection	Time (MYA)
*SlPLATZ2* vs. *SlPLATZ3*	0.1130	0.2189	0.5162	tandem	Purifying selection	7.30
*SlPLATZ3* vs. *SlPLATZ4*	0.0279	0.0315	0.8857	tandem	Purifying selection	1.05
*SlPLATZ6* vs. *SlPLATZ7*	0.0414	0.0468	0.8855	tandem	Purifying selection	1.56
*SlPLATZ15* vs. *SlPLATZ16*	0.0218	0.0704	0.3092	tandem	Purifying selection	2.35
*SlPLATZ13* vs. *SlPLATZ18*	0.0948	0.7982	0.1188	Segmental	Purifying selection	26.61
*SlPLATZ14* vs. *SlPLATZ20*	0.1004	0.8476	0.1184	Segmental	Purifying selection	28.25
*SlPLATZ17* vs. *SlPLATZ18*	0.1477	0.9430	0.1567	Segmental	Purifying selection	31.43

Ks, the number of synonymous substitutions per synonymous site; Ka, the number of non-synonymous substitutions per nonsynonymous site; MYA, million years ago.

## Data Availability

Not applicable.

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
