# Peer review of "Comprehensive Genome-Wide Analysis and Expression Pattern Profiling of PLATZ Gene Family Members in Solanum Lycopersicum L. under Multiple Abiotic Stresses"

_plants, 2022, doi:10.3390/plants11223112_

Round 1
Reviewer 1 Report
My comments are below:
English editing is required. It would appear that this paper was not peer-reviewed for English language use.
Inconsistencies in tables and their legends.
Abstract
Abstract is carelessly written authors should incorporate their notable findings and adequately connect with the sentences they choose to correspond.
Introduction
The introduction section must have a clear hypothesis and significantly develop the second paragraph of your manuscript. Make it more connected to the problem statement.
Overall, there is the repetition of the information, which could be avoided.
Discussion
This section should include more information and references related to the relevant and related works.
Conclusions
If possible, restructure and carefully edit the conclusion section and add clear information regarding the most noteworthy findings.
Author Response
Please, find the attached file.

Reviewer 2 Report
The manuscript by Wai et al. entitled „ Comprehensive genome-wide analysis and expression pattern profiling of PLATZ gene family members in Solanum lycopersicum L. under multiple abiotic stresses“ presents essential new data about transcription factors PLATZ gene family.
Materials and methods used in this study have been sufficiently described. It is worth emphasizing the authors' s analyse importance of both cis‑acting elements and miRNA target sites in PLATZ genes expression.
Overall, this is an interesting manuscript that presents the findings of a well designed and executed research.
Some minor remarks:
- line 110 „In silico“ should be italic
- lines 143-147 and in other places. Species and genus names should be written in italic.
- lines 203-205 and in other places. Genes should be written in italic.
- line 436 „pathogen Phytophthora capsic“ should be Phytophthora capsici
- line 757-758 The 2−ΔΔCt method was employed to analyze the relative expression of each gene against each treatment [83]. Correct reference is [84].
Author Response
Please, find the attached file.

Reviewer 3 Report
This article aimed to examine Comprehensive genome-wide analysis and expression pattern profiling of PLATZ gene family members in Solanum lycopersicum L. under multiple abiotic stresses. Before recommending this article for publication, there are some shortcomings for that should be resolve.
Abstract
Add comma after “DNA-binding motifs” in the 2nd line of the abstract.
Write full form at first use “SlPLATZ”
Specify processes and ligands “roles in several metabolic processes and their binding affinity for various ligands”
Introduction
The introduction part is well written but still some details are required.
Gene family names and plant names must be italicized in the whole text.
Add information about the significance of genome wide studies in the first para.
Line 55 could be cited with recent study https://doi.org/10.1007/s10725-021-00785-7
Provide information and economic importance of the tomato.
Line 96 “conducted” should be cited with relevant studies.
Results
Figure 2 legend italicize the species names.
Methods
Line 730 to 732 could be cited with relevant study.
https://doi.org/10.3390/ijms22179175.
Conclusion
Conclusion is well justified.
Author Response
Please, find the attached file.

Round 2
Reviewer 1 Report
Manuscript can be accepted for publication.